# In vivo treatment with a non-aromatizable androgen rapidly alters the ovarian transcriptome of previtellogenic secondary growth coho salmon (*Onchorhynchus kisutch*)

**Christopher Monson** [1], **Giles Goetz**[2], **Kristy Forsgren**[3], **Penny Swanson**[2,4], **Graham Young**[1,4]*

1 School or Aquatic and Fishery Sciences, University of Washington, Seattle, Washington, United States of America, 2 Northwest Fisheries Science Center, National Marine Fisheries Service, National Oceanographic and Atmospheric Administration, Seattle, Washington, United States of America, 3 Department of Biological Science, California State University, Fullerton, Fullerton, California, United States of America, 4 Center for Reproductive Biology, Washington State University, Pullman, Washington, United States of America

* grahamy@uw.edu

## Abstract

Recent evidence suggests that androgens are a potent driver of growth during late the primary stage of ovarian follicle development in teleosts. We have previously shown that the non-aromatizable androgen, 11-ketotestosterone (11-KT), both advances ovarian follicle growth in vivo and dramatically alters the primary growth ovarian transcriptome in coho salmon. Many of the transcriptomic changes pointed towards 11-KT driving process associated with the transition to a secondary growth phenotype. In the current study, we implanted previtellogenic early secondary growth coho salmon with cholesterol pellets containing 11-KT and performed RNA-Seq on ovarian tissue after 3 days in order to identify alterations to the ovarian transcriptome in early secondary growth. We identified 8,707 contiguous sequences (contigs) that were differentially expressed (DE) between control and 11-KT implanted fish and were able to collapse those to 3,853 gene-level IDs, more than a 3-fold more DE contigs than at the primary growth stage we reported previously. These contigs included genes encoding proteins involved in steroidogenesis, vitellogenin and lipid uptake, follicle stimulating hormone signaling, growth factor signaling, and structural proteins, suggesting androgens continue to promote previtellogenic secondary growth.

## Introduction

In fish, the development of a competent oocyte is divided into three general stages: primary growth, secondary growth, and maturation [1,2]. These stages are under the control of numerous endocrine and paracrine factors [1], and recent studies using teleost models indicate that androgenic and estrogenic steroids play stage-specific roles in regulating development [2].

During the perinucleolar stage of primary growth, exposure to a non-aromatizable androgen, 11-ketotestosterone [11-KT] was effective in increasing the volume of ovarian follicles of

**Data Availability Statement:** All relevant data are within the manuscript and its Supporting information files.

**Funding:** This study was supported by National Science Foundation awards NSF-OISE-0914009 awarded to KF, NSF IOS-0940765 awarded to GY and NSF IOS-1921746 awarded to GY. NSF.gov. The funders had no role in study design, data collection and analysis, decision to publish, or preparation of the manuscript.

**Competing interests:** The authors have declared that no competing interests exist.

coho salmon [3,4] and Atlantic cod [5]. Treatment with estradiol-17β [E2] was less effective in coho salmon; 20 days of in vivo exposure to E2 was required before a significant increase in volume occurred [6], in contrast to 10 days of treatment with 11-KT [4]. These results suggest that androgen signaling may be a primary steroidal driver of growth at this stage. Conversely, in vivo [6,7] or vitro [3] treatment with E2 promoted the formation of cortical alveoli, an indicator of secondary follicle development. This effect was absent in 11-KT treated follicles during primary growth.

In a previous study using deep transcriptome sequencing, hundreds of ovarian follicle transcripts in which expression levels were altered by in vivo 11-KT treatment were identified, prior to development of the secondary follicle phenotype [4]. These included transcripts encoding proteins involved in steroidogenesis and steroid action, growth factor signaling, and the extracellular matrix. Pathway analysis identified biological functions and canonical pathways that were potentially altered, including ovarian development, tissue differentiation and remodeling, and lipid metabolism. Plasma E2 levels were also increased by this treatment, as well as *fsh* transcript levels, both hallmarks of entry into secondary growth [8,9]. Together, these results suggest that androgens promote both primary ovarian follicle development, and the transition into secondary ovarian follicle growth.

Secondary growth is characterized by the activation of the brain-pituitary-gonads axis that results in an increase in ovarian E2 synthesis via Fsh signaling. The presence of cortical alveoli is a histological indicator of entry into secondary growth in coho salmon [8]. Several lines of evidence implicate androgens in early secondary ovarian follicle development as well. 11-KT promotes the accumulation of lipids and an increase in size of previtellogenic eel follicles [10–12]. In early secondary coho salmon follicles, in vitro treatment with 11-KT was as effective as E2 in increasing the size of ovarian follicles, although E2 caused a much greater increase in cortical alveoli [3]. While the growth and cytological effects of these steroids (androgens in particular) on follicle growth have been characterized in several species, there are fewer data on mechanisms driving them. In order to identify the mechanisms underlying the growth-promoting actions of androgens in early secondary growth, we implanted female coho salmon containing ovaries at the cortical alveolus stage with sustained release pellets containing 11-KT. Changes in the ovarian follicle transcriptome were determined using RNA-Seq followed by pathway analysis after three days.

## Methods

### Chemicals and general animal procedures

11-Ketotestosterone was purchased from Steraloids (Newport, RI). Cholesterol was purchased from Sigma-Aldrich (St. Louis, MO). L-15, hematoxylin, eosin, and diethyl ether were purchased from Thermo Fisher Scientific (Waltham, MA). Bouin's fixative was purchased from Ricca Chemical Company (Arlington, TX).

Juvenile coho salmon (Issaquah Hatchery stock, Issaquah, WA) were reared at the hatchery facilities of the Northwest Fisheries Science Center, Seattle, WA under simulated natural photoperiod in re-circulated 10–11˚C fresh water, under an approved protocol according to guidelines established by the Institutional Animal Care and Use Committee, University of Washington (protocol 4078–04). Fish were fed twice daily with a commercial feed (BioDiet, Bio-Oregon, Longview, WA) according to the manufacturer's guidelines.

Genetic sex was determined using an established sex marker in tagged fish. Genetic females were segregated into an all-female stock which was reared as previously described [4]. At the termination of experiments, fish were anesthetized in buffered 0.05% tricaine methanesulfonate until movement of the gill operculum ceased. Fork length and body weight were

measured. Blood was collected from the caudal vein and immediately transferred to heparinized microcentrifuge tubes and placed on ice. Blood plasma was separated by centrifugation at $1200 \times g$ for 15 minutes. After decapitation, ovaries were removed and weighed and then either snap frozen in liquid nitrogen or fixed in Bouin's fixative for histological analysis.

## Experimental procedures

Female juvenile coho salmon (121±2.8 g, 2-years of age) were implanted with either blank cholesterol pellets or cholesterol pellets containing 10 μg 11-KT. The amount of steroid included in the pellets was determined in preliminary experiments to result in significant but physiologically relevant increases in plasma steroid levels. As in our previous study [4], pellets containing 11-KT were incubated for 24 hours in sterile L-15 media (Thermo Fisher Scientific) and were then washed with L-15 prior to implantation to temper the initial release rate. Fish were lethally sampled after 3 days. At that time, fork length, body weight, and gonad weight were measured, and blood and ovaries were collected as described above. Histological screening eliminated any female that displayed overtly asynchronous ovarian stages or were not at the cortical alveolus stage, and frozen ovarian samples from control, and 11-KT treated females (N = 3 per group) were selected for RNA-Seq analysis.

## Sex steroid assays

Steroids were double extracted from 250 μl of plasma using diethyl ether (1.5 ml x 2) and extracts were evaporated under a nitrogen gas stream, then re-suspended in appropriate buffers as previously described [4]. Plasma 11-KT levels were measured by enzyme-linked immunoassay [13] using tracer and secondary antibody coated plates from Cayman Chemicals (Ann Arbor, MI) and primary antibody donated by David Kime (University of Sheffield, UK). Plasma E2 was measured by radioimmunoassay, as described by Sower and Schreck [14], and modified by Fitzpatrick et al. [15].

## Histological analysis

Fixed ovarian tissues were washed with 70% ethanol, dehydrated in increasing concentrations of ethanol and xylene, and embedded in paraffin wax. Sections with a thickness of 5 μm were cut and mounted on microscope slides and stained with hematoxylin and eosin. Average ovarian follicle volume was calculated from at least 15 follicles per sample, measuring follicles that were sectioned through the nucleus of the oocyte with an image analysis system (NIS-elements, Nikon, USA), as described previously [3], and oocytes were scored for stage based on previously published criteria [3,8].

## RNA extraction

Total RNA was extracted using Qiagen RNEasy mini kit (Qiagen, Hilden, Germany) according to the manufacturer's guidelines. RNA pellets were re-suspended in DNAse/RNAse free water (Sigma-Aldrich). Total RNA concentrations in extracts were determined using a NanoDrop ND-100 (NanoDrop Technologies, Wilmington, DE).

## RNA-Seq and pathway analysis of alteration in the ovarian transcriptome

**i. Sample preparation.** Total ovarian RNA (200 ng) was submitted to Omega Bio-Tek Inc (Norcross, GA) for quality checking, library preparation (poly A selected), and 100 base pair, paired-end sequencing.

**ii. RNA-seq analysis.** Bioinformatic analyses were performed using the DRAP pipeline as previously described [4,16]. Briefly, sequences were quality trimmed using Trim Galore v0.4.0 [17] and assembled into a de novo backbone with Drap v1.8 [16] and Oases v0.2.09 [18] using the kmer values of 19, 23, 25, 27, 31, and 35. Contiguous sequences (contigs) that had FPKM (fragments per kilobase of transcript per million mapped reads) greater than 1 and had sequence lengths greater than 200 bp were retained. These contigs were annotated using BlastX against the NCBI non-redundant protein database (nr) and partially non-redundant nucleo-tide database (nt); only sequences with an E-value ≤-05 were retained. Gene level count esti-mates were made using RSEM v.1.2.31 [19] and bowtie2 v2.2.6 [20] and differential expression was determined using DESeq2 [21]. Contigs with a P-adjusted (P-adj) value ≤0.1 were consid-ered significantly altered between control and 11-K. To control for sequencing errors and dif-ferences in sequencing depth leading to misidentification of differential expression of contigs with low read counts, those contigs with a basemean ≤10 were excluded from further analysis. Gene clustering was performed using cluster::agnes package in R with the Spearman method [22]; data were $\log_2$ transformed and centered on a mean expression value to improve visuali-zation of expression differences.

**iii. Pathway analysis.** Ingenuity Pathway Analysis® (IPA) software was used to conduct pathway and network analyses and predict the effects of steroid treatment on biological func-tions. Contigs were initially mapped to zebrafish orthologs using BLASTN against the Ensembl *Danio rerio* gene database (v.Zv9.72). However, some zebrafish genes have not been mapped to mammalian orthologs, so the remaining contigs were mapped to the *Homo sapiens* transcript database (v.GRCH37.72) for inclusion in IPA. If more than one contig (P≤0.05) mapped to the same gene, the average expression value of those contigs was used as the gene expression value in further analyses. The expression patterns of the zebrafish and human gene orthologs were compared to the IPA database to estimate altered canonical pathways and biological func-tions (Fisher exact test P≤0.05 [$-\log_{10}$ P-value ≥1.3]). This program generates networks that maximize the connectivity of genes with significantly altered expression based on known func-tional interactions [23], predicts alterations in biological function, and predicts both upstream and downstream regulators given the direction of expression differences in given gene sets. A z-score was calculated to identify predicted increases or decreases in biological functions in treated samples relative to controls. The z-score is a statistical measure of the match between expected and observed gene expression direction. Zebrafish nomenclature is used throughout when referring to fish species, although due to the use of this software, human gene names are used in places where annotation to the zebrafish database was not possible.

## Results

### Morphometrics and sex steroid levels

No significant treatment effects on fish selected for analysis were observed between control and treated samples with regard to fork length (21.6±0.2 cm), body weight (121.6±2.8 g), GSI (0.4±0.02), ovarian follicle volume (0.044±0.004 mm$^2$), or gross ovarian follicle morphology. All samples selected for sequencing displayed an early to mid-cortical alveolus stage phenotype (Fig 1A–1C). The mean plasma 11-KT level in control fish was 0.12±0.04 ng/ml. In samples selected for RNA-Seq, treatment with 11-KT for 3 days significantly increased plasma 11-KT levels to 18.5±3.7 ng/ml (Fig 1D) but did not alter E2 levels (0.2±0.1 ng/ml).

### RNA-seq

Sequencing resulted in 1.6 billion total reads from 9 samples (Table 1). Following quality trim-ming and pairing, greater than 99.7% of reads were retained. De novo assembly generated

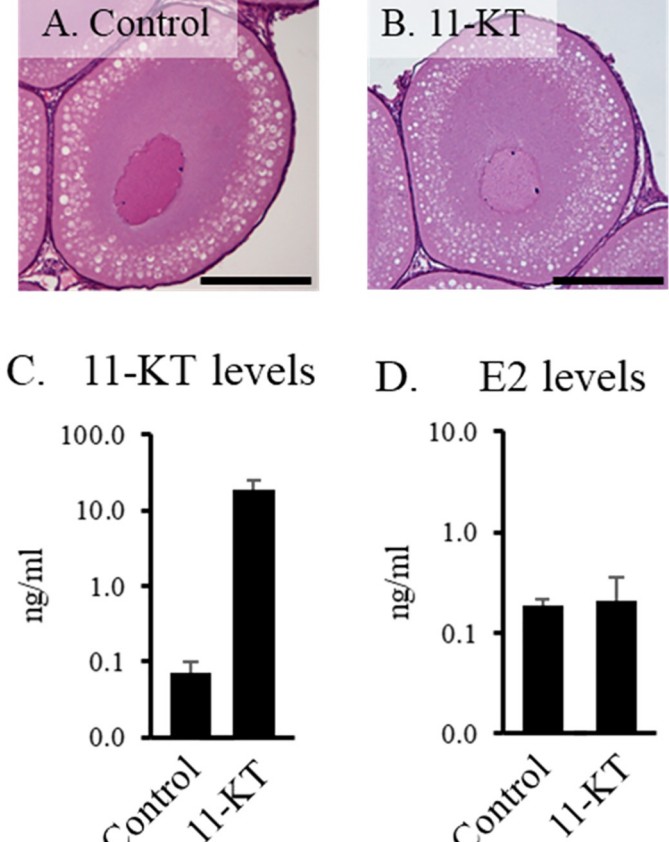

**Fig 1. Ovarian stage and plasma sex steroid levels.** Ovarian tissue from fish displaying early secondary growth ovarian follicles with similar cortical alveoli abundance were chosen from each treatment, (A) control, (B) 11-KT, for RNA-seq analysis. Scale bar = 200μm. Plasma 11-KT (C) and E2 (D) levels were measured three days after implant in control and 11-KT treated samples (N = 3). Asterisks indicate significant elevation in plasma steroid levels (P <0.05).

**Table 1. Summary statistics of the RNA-Seq pipeline.**

| Treatment | Sample | Raw Reads | Paired Reads | Trimmed Reads | Mapped Reads | Percent Mapped |
|---|---|---|---|---|---|---|
| Control | 1 | 164.464,526 | 82,232,263 | 82,071,323 | 66,591,651 | 81.14 |
| | 2 | 214,106,184 | 10,753,092 | 106,790,936 | 86,663,462 | 81.15 |
| | 3 | 170,020,428 | 85,005,214 | 84,830,356 | 67,743,146 | 79.86 |
| | Average | *182,860,379* | *91,430,190* | *91,230,872* | *77,203,304* | *80.72* |
| 11-KT Treated | 1 | 174,052,222 | 87,026,111 | 86,854,547 | 71,104,279 | 81.87 |
| | 2 | 170,960,500 | 58,480,250 | 85,297,177 | 69,699,048 | 81.71 |
| | 3 | 168,526,044 | 84,263,022 | 84,100,132 | 67,897,057 | 80.73 |
| | Average | *171,179,589* | *85,589,794* | *85,417,285* | *69,566,795* | *81.44* |
| Additional reads included in backbone | 1 | 190,527,516 | 95,263,758 | 95,048,317 | 76,101,066 | 79.97 |
| | 2 | 168,924,010 | 84,462,055 | 84,209,920 | 68,399,236 | 81.22 |
| | 3 | 175,561,102 | 87,780,551 | 87,606,079 | 71,701,627 | 81.85 |
| | Average | *178,337,543* | *89,168,771* | *88,954,772* | *72,036,976* | *81.01* |
| | **Total** | **1,597,132,532** | **798,566,266** | **796,808,787** | **579,217,921** | **Avg. = 81.06** |

Number of raw reads, number of paired reads, number after quality trimming, number mapped, and percent of reads mapped for Control, 11-KT treated, and additional reads included in creation of backbone but not used in the differential expression analyses.

63,423 contigs between 201 bp and 15,353 bp with a mean contig length of 1,673 bp. Eighty-one percent of reads were mapped to the de novo backbone. A total of 63,048 of these contigs (99.4%) were annotatable.

RNA-seq analysis identified 8,707 contigs that were differentially expressed (DESeq2, P-adjusted ≤0.1) from controls in 11-KT (Fig 2A). Cluster analysis of differentially expressed contigs in 11-KT treated samples (Fig 2B) demonstrated distinct differences in expression patterns between control and treatment groups.

## Alterations in the ovarian transcriptome induced by short-term 11-KT treatment

Of the 63,423 contigs generated, 43,820 (69%) were annotated to zebrafish or human gene orthologs and mapped to IPA. Duplicate contigs were collapsed to the gene-ID level using the

A.

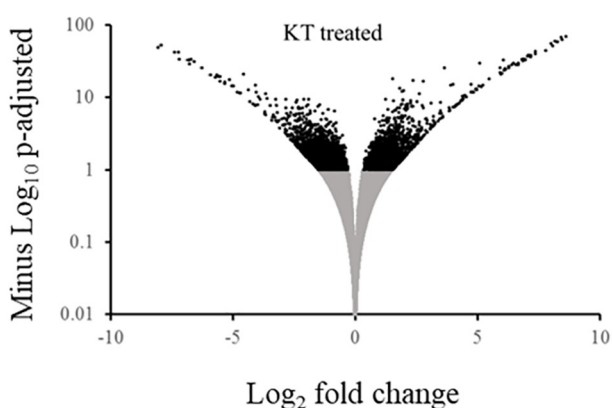

B.

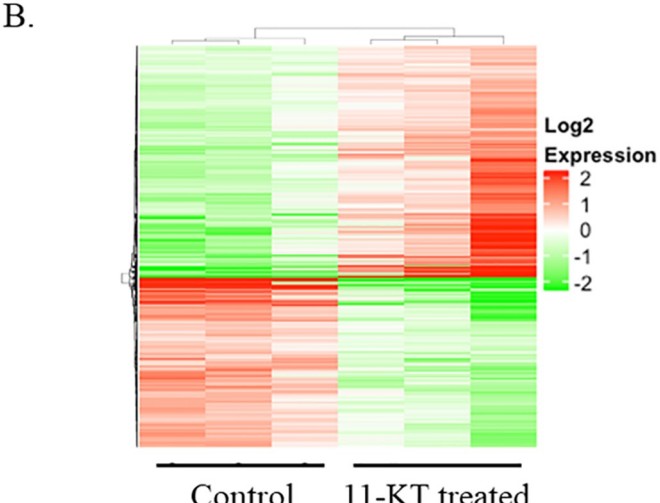

**Fig 2. The expression of contigs in ovaries from females following short term treatment with 11-KT.** All contigs with a calculated p-adjusted value (-Log$_{10}$), plotted by fold change (Log$_2$). Black dots represent contigs significantly altered by 11-KT (A). Cluster analysis (DESeq2) of differentially expressed contigs (DESeq2, basemean>10, P-adj<0.1) after three days of 11-KT treatment (B). Expression of contigs (rows) is displayed for three independent samples (columns), with red representing up-regulation and green representing down-regulation from the mean expression value (white) of each contig. Each column represents data from ovaries of a single individual.

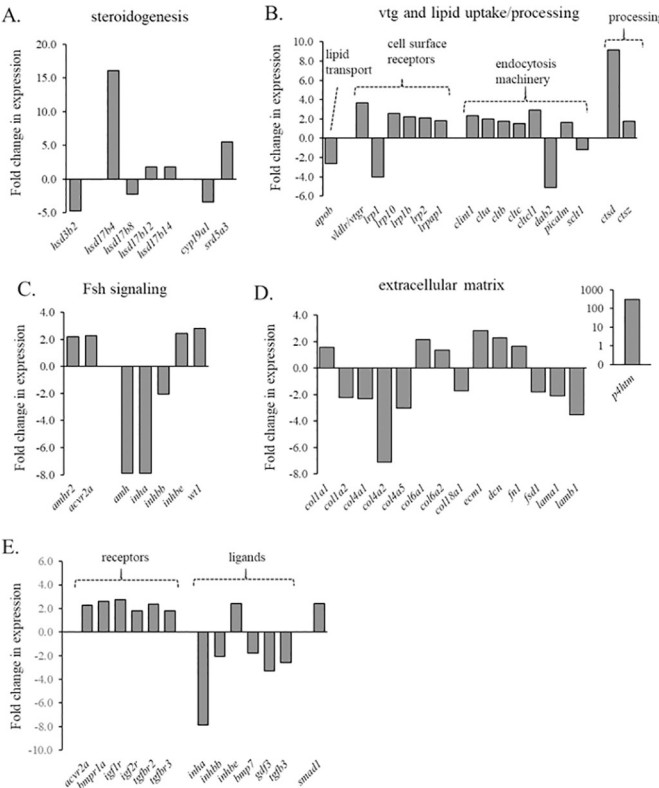

**Fig 3. The expression of contigs altered by 11-KT treatment related to the morphology and function of the ovarian follicle.** The fold expression compared to controls is displayed for contigs mapped to genes involved in (A) steroidogenesis, (B) vitellogenesis and lipid uptake and processing, (C) Fsh signaling, (D) the extracellular matrix, and (E) *tgfb* superfamily members.

average expression value relative to controls, resulting in 3,853 genes with expression altered by 11-KT (P-adjusted ≤0.1).

Exposure to 11-KT altered the expression of genes that encode proteins involved in steroid synthesis or metabolism (Fig 3A), proteins involved in vitellogenin and lipid uptake and processing (Fig 3B), proteins that mediate Fsh signaling or expression (Fig 3C), extracellular matrix proteins (Fig 3D), and growth factors (Fig 3E).

## Pathway analysis

IPA software was used to identify canonical pathways and biological functions altered by 3 days of 11-KT exposure. A total of 263 and 12 canonical pathways were significantly associated ($-Log_{10}$ P ≥ 1.31) with 11-KT (Table 2) treatment. A total of 49 canonical pathways were predicted to be significantly altered by the IPA z-score algorithm following 11-KT treatment. Of these, two of the pathways most significantly associated with our dataset that had significant z-scores are involved in cell adhesion to the extracellular matrix: integrin signaling ($-Log_{10}$ P = 13.00, z-score = 3.53) and actin cytoskeleton signaling ($-Log_{10}$ P = 8.87, z-score = 2.26). Additional pathways significantly associated with 11-KT treatment included insulin receptor signaling ($-Log_{10}$ p = 9.40), estrogen receptor signaling ($-Log_{10}$ P = 7.75), androgen signaling ($-Log_{10}$ P = 7.12), GnRH signaling ($-Log_{10}$ P = 5.05), and clathrin-mediated endocytosis signaling ($-Log_{10}$ P = 6.89).

**Table 2. Canonical pathways identified in ovaries of females treated with 11-KT for three days, identified by Ingenuity® pathway analysis software.**

| Ingenuity Canonical Pathways altered by 11-KT at day 3 | -Log (P-value) | z-score | ratio of genes |
|---|---|---|---|
| Germ Cell-Sertoli Cell Junction Signaling | 13.00 | | 73/173 |
| Integrin Signaling | 13.00 | 3.53 | 86/219 |
| Remodeling of Epithelial Adherens Junctions | 10.80 | 0.89 | 37/68 |
| Rac Signaling | 10.40 | 0.57 | 52/117 |
| Epithelial Adherens Junction Signaling | 10.20 | | 60/146 |
| Sertoli Cell-Sertoli Cell Junction Signaling | 10.20 | | 69/178 |
| Molecular Mechanisms of Cancer | 9.54 | | 117/374 |
| Insulin Receptor Signaling | 9.40 | 0.94 | 57/141 |
| Signaling by Rho Family GTPases | 9.32 | 1.73 | 85/247 |
| Actin Cytoskeleton Signaling | 8.87 | 2.26 | 79/228 |
| Pyridoxal 5'-phosphate Salvage Pathway | 8.68 | | 33/65 |
| Breast Cancer Regulation by Stathmin1 | 8.64 | | 74/208 |
| Neuregulin Signaling | 8.54 | 3.18 | 40/88 |
| Phagosome Maturation | 8.51 | | 56/144 |
| Tight Junction Signaling | 8.41 | | 62/167 |
| PI3K/AKT Signaling | 8.29 | 1.18 | 50/124 |
| RhoA Signaling | 8.07 | 2.24 | 49/122 |
| Huntington's Disease Signaling | 8.00 | 0.91 | 80/241 |
| mTOR Signaling | 7.85 | 1.66 | 69/199 |
| NGF Signaling | 7.78 | 1.94 | 47/117 |
| Estrogen Receptor Signaling | 7.75 | | 50/128 |
| RhoGDI Signaling | 7.73 | -2.45 | 62/173 |
| AMPK Signaling | 7.68 | 0.58 | 66/189 |
| Glucocorticoid Receptor Signaling | 7.56 | | 90/287 |
| 14-3-3-mediated Signaling | 7.49 | 0.71 | 50/130 |
| ILK Signaling | 7.37 | 1.78 | 67/196 |
| Ephrin Receptor Signaling | 7.22 | 1.70 | 61/174 |
| Mitotic Roles of Polo-Like Kinase | 7.18 | 0.26 | 31/66 |
| Fcγ Receptor-mediated Phagocytosis in Macrophages and Monocytes | 7.16 | 2.40 | 39/93 |
| Protein Ubiquitination Pathway | 7.14 | | 81/255 |
| Androgen Signaling | 7.12 | 0.66 | 46/116 |
| Prostate Cancer Signaling | 7.01 | | 39/94 |
| Clathrin-mediated Endocytosis Signaling | 6.89 | | 66/197 |
| ERK/MAPK Signaling | 6.71 | 2.39 | 66/199 |
| HIPPO signaling | 6.64 | 0.00 | 36/86 |
| Regulation of eIF4 and p70S6K Signaling | 6.57 | 2.12 | 55/157 |
| NRF2-mediated Oxidative Stress Response | 6.52 | 2.96 | 64/193 |
| TGF-β Signaling | 6.49 | 0.93 | 36/87 |
| Superpathway of Inositol Phosphate Compounds | 6.49 | | 73/230 |
| Regulation of Actin-based Motility by Rho | 6.42 | 1.86 | 37/91 |
| IGF-1 Signaling | 6.35 | 2.54 | 41/106 |
| Reelin Signaling in Neurons | 6.28 | | 37/92 |
| Gap Junction Signaling | 6.25 | | 57/168 |
| Telomerase Signaling | 6.18 | 1.18 | 42/111 |
| RAR Activation | 6.07 | | 62/190 |

Pathway, -Log$_{10}$ P-value, z-score, ratio of genes represented in data set to total genes reported in pathway. Top 45 pathways displayed. Gray shaded rows indicate pathways predicted to be significantly altered by 11-KT treatment.

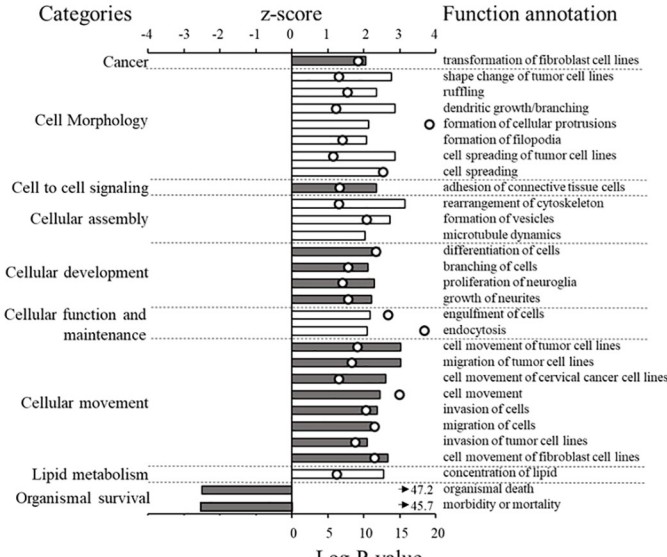

**Fig 4. Biological functions in the ovary altered by 11-KT treatment.** Many biological functions were predicted to be altered in response to 11-KT treatment by IPA pathway analysis, including cellular functions, movement, and morphology, as well as lipid metabolism. Functions are predicted to be activated or inactivated based on a positive or negative z-score (top axis, $\geq 2$ or $\leq$-2 is considered significantly predictive). The P-value (white dots, lower axis, -$\log_{10}$ P $\geq 1.3$ [P $\leq 0.05$]) indicates the likelihood that a function is accurately associated with the genes in our data set. Arrows indicate—$\log_{10}$ P-value is greater than the bounds of the axis. Bar color indicates biological functions within the same category.

IPA was used to predict changes in biological functions following 11-KT exposure. 11-KT treatment led to the predicted significant alteration ($|z| \geq 2$) of 26 biological functions in the ovary. Many of these biological functions in the ovary of 11-KT treated fish were in categories related to cellular processes (Fig 4), but also to lipid metabolism and organismal survival.

## Discussion

Our previous studies on coho salmon have shown that low concentrations of 11-KT induce growth and development of primary ovarian follicles in vitro and in vivo [6], and dramatically alter the ovarian transcriptome [4]. Both E2 and 11-KT are potent stimulators of early secondary follicle growth [3,6]. In the present study, we used the previously described in vivo steroid exposure model to undertake deep transcriptome sequencing of ovarian tissue in order to identify early transcriptional changes resulting from 11-KT exposure during previtellogenic secondary growth.

After three days of sex steroid exposure, 11-KT dramatically altered the ovarian transcriptome. These widespread transcriptomic changes induced by 11-KT are consistent with a role for 11-KT in lipid and vitellogenin uptake and processing, and in Fsh signaling, which are hallmarks of secondary growth [11], as well as cellular development and other cellular processes, and changes in the extracellular matrix.

### The effects of 11-KT on early secondary ovarian follicle transcriptome

In our previous study [4], we exposed female coho salmon in the late perinucleolar stage of primary growth to 11-KT for 1 and 3 days, and performed RNA-Seq and pathway analyses on ovarian tissue. We identified numerous differentially expressed genes that encode proteins

involved in steroidogenesis and steroid action, including follicle stimulating hormone receptor (*fshr*). These results implicate androgens in processes that prepare the ovarian follicle for secondary growth (i.e., Fsh-mediated E2 synthesis). Additionally, we identified canonical pathways that indicated potential modifications to the extracellular matrix and potential alterations in biological functions involved in reproductive development. These results led to the hypothesis that 11-KT plays a major role in primary growth, enhancing the potential for Fsh- and E2-mediated signaling in secondary growth. Consistent with the potent growth-promoting effects of 11-KT on early secondary follicles in vitro [3], the large number of contigs (8,707) and corresponding genes (3,853) that were differentially expressed, and the magnitude of the fold change in many gene transcripts after 3 days of 11-KT treatment in the current study indicate that the early secondary ovarian follicle is even more sensitive to androgen signaling.

The fundamental difference between the previous study and the current RNA-Seq experiment was the ovarian follicle stage, as identified by histological indices (absence/presence of cortical alveoli). The ovarian follicles in the present study contained peripheral cortical alveoli consistent with the morphology of the cortical alveolus stage of early secondary growth. However, common themes did emerge from the biological functions predicted to be altered after 3-days of 11-KT treatment in the two studies (Fig 5). Biological functions in categories of cell-to-cell signaling and interaction, cellular development, and cellular movement were activated (z-score >2) in both studies. The biological functions of morbidity and mortality and organismal death were predicted to be significantly inhibited (z-score <-2) in both studies. These predictions point towards 11-KT playing a similar role in regulating basic cellular processes and cell survival at these stages.

Notable follicle stage-associated differences in the transcriptomic response to 11-KT include genes encoding proteins involved in steroid synthesis and the synthesis of E2 (*hsd3b*,

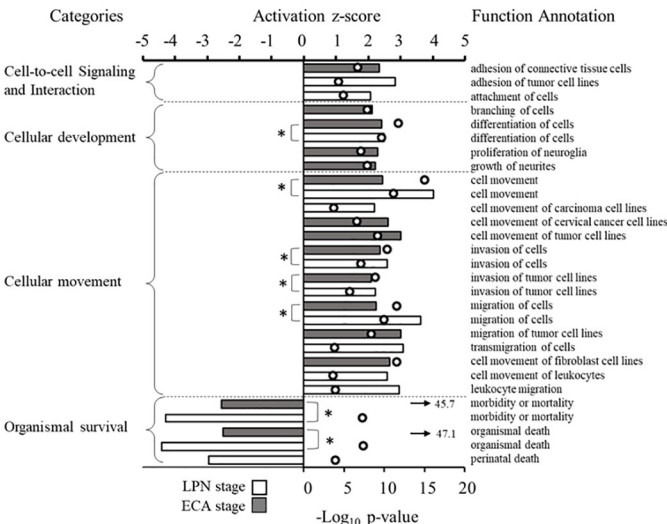

**Fig 5. Altered biological functions in the ovary from categories in common between 11-KT treated primary and early secondary growth coho salmon.** Many of the same processes were predicted to be altered by 11-KT after 3-days at both late primary growth (white bars) and early secondary growth (gray bars) stages. A previously published DEG list [4] was interrogated using the IPA biological function analysis and compared with the biological functions predicted to be altered in the current study. A positive or negative z-score (top axis, $\geq$2 or $\leq$-2 is considered significantly predictive). The P-value (white dots, lower axis-Log$_{10}$ P $\geq$1.3 [P $\leq$0.05]) indicates the likelihood that a function is accurately associated with the genes in our data set. Arrows indicate—Log$_{10}$ P-value is greater than the bounds of the axis. Starred functions were significantly altered at both stages.

*cyp19a1*), even though very similar 11-KT levels were achieved with implants in each RNA-Seq study. The expression of these genes was significantly increased by 11-KT in primary follicles [4], but was significantly decreased by 11-KT in early secondary follicles. The products of these genes both act to catalyze the production of sex steroids, and specifically, *cyp19a1* encodes aromatase, the enzyme responsible for the conversion of testosterone (T) to E2. Decreases in expression indicate that the potential for production of E2 by previtellogenic secondary follicles was reduced by 11-KT, potentially due to the actions of 11-KT at different levels of the brain-pituitary-ovary axis. 11-KT also altered the expression of several hsd17 genes, with a notable 15-fold increase in hsd17b4. Multiple hsd17 genes have diverse catalytic functions, including the interconversion of high-activity, 17β-hydroxyl, and low-activity, 17-keto, forms of C-19 (androgen) and C18 (estrogen) steroids [24]. Transcripts encoding another steroidogenic enzyme, Srd5a3 were upregulated 5.48–fold after 11-KT exposure. The 5α-reductase activity of Srd5a3 catalyzes the conversion from T to DHT [25] or potentially promotes conversion of 11-KT to other potent non-aromatizable androgens [26]. Taken together, these results suggest that in early secondary growth, 11-KT alters steroidogenic capacity of the ovarian follicle, and in particular, increases the potential for the production of 5α-reduced androgens.

## Genes and pathways characteristic of the secondary follicle

Given the abundance of differentially expressed genes following 11-KT treatment, we focused on further analyses on genes and pathways potentially involved in processes characteristic of early secondary follicle development: (i) vitellogenin (Vtg) and lipid uptake; (ii) Fsh signaling; (iii) structural changes in the ovarian follicle; and (iv) changes in growth factor signaling.

**i. 11-KT effects on gene transcripts involved in vitellogenesis and lipid uptake.**   After 3 days of treatment with 11-KT, the expression of very low-density lipoprotein receptor/vitellogenin receptor (*vldlr/vtgr*), cathepsin d (*ctsd*), cathepsin z (*ctsz*), and clathrin light chain a (*clta*), clathrin light chain b (*cltb*), and clathrin heavy chain (*cltc*) was increased. The proteins encoded by these genes play fundamental roles in oocyte development, controlling the uptake of lipids and Vtg during vitellogenesis [2,27].

Teleost *vtgr* is a splice variant of *vldlr*, only lacking an *O*-linked sugar domain, and the sequence of the contig mapped to *vtgr/vldlr* in our dataset does not cover that domain. The contig mapped with 98% identity to rainbow trout *vtgr* (LOC100136065). In rainbow trout, a closely related species in the same genus as coho salmon, Vtgr appears to specifically bind Vtg, whereas additional somatic lipoprotein receptors bind very low-density lipoprotein (Vldl), and low-density lipoprotein (Ldl) [28].

Most of what is known about the effects of androgens on vitellogenesis or Vtg production is from studies linking androgen exposure to increases [29,30] or decreases [31] in hepatic Vtg production. The expression of *vtgr* has been reported for oocytes of several species and these studies indicate collectively that transcript levels peak during primary growth and decline thereafter [32–37]. Transcripts may also be stored and later translated during vitellogenesis [38]. Conversely, the endocrine or paracrine control of *vtgr* expression by sex steroids has only been described in several species, largemouth bass [39] and medaka [40]. E2 was shown to repress *vtgr* expression [41] and insulin (Ins) increased *vtgr* expression in previtellogenic follicles [39], although co-exposure with Ins and either 11-KT or E2 reduced the Ins-induced expression in vitro [39], suggesting that androgen and estrogen receptor signaling may regulate insulin-mediated pathways [2]. The latter study used very high concentrations of 11-KT or E2 (500 nM; approximately 151 ng/ml or 135 ng/ml respectively) which is >8-fold higher than the plasma concentration we achieved with our implants (18.5 ng/ml) and may explain the

differences in expression following 11-KT exposure in our model. Interestingly, in the present study, the expression of both ovarian *ins* and *insulin receptor* (*insr*) was reduced with 11-KT treatment.

The Vtgr protein is active at the oocyte cell surface during vitellogenesis and has been associated with endocytotic clathrin-coated pits. Following endocytosis of the Vtg-Vtgr complex, Vtg is then cleaved into component yolk proteins by lysosomal cathepsins (Cts), which recognize particular amino acid sequences in the Vtg protein. Transcripts for *ctsz* were expressed in early vitellogenic follicles in mummichog [42] and throughout vitellogenesis in carp [43], and increased by 11-KT in this study. In teleosts, Ctsd has been implicated in the cleavage of Vtg into the three primary yolk components, lipovitellin, phosvitin, and the β'-component [44]. In coho salmon ovarian follicles, the expression of *ctsb* was inversely correlated with the transition to secondary growth, whereas the expression of *ctsd* and *ctsz* was unchanged between late primary and early secondary growth [35]. However, ovarian cathepsins likely undergo post-transcriptional regulation, and thus transcript levels may not correlate well with enzymatic activity [44].

A significant characteristic of the transition to secondary growth is accumulation of neutral lipids in the ooplasm of the oocyte. In the present study, 11-KT altered the expression of lipid transfer genes in secondary stage follicles. 11-KT decreased expression of *apob*, a member of the large lipid transfer protein superfamily that includes Vtg. The protein encoded by *apob* is the primary protein component of Vldl and Ldl molecules. The expression of Ldl receptor-related proteins *lrp1* (decreased 4.04 fold), *lrp1b* (increased 2.21 fold), *lrp2* (increased 2.11 fold), *lrp5* (decreased 1.62 fold), and *lrp10* (increased 2.59 fold), and Ldl receptor-related protein associated protein 1 (*lrpap1*, increased 1.83 fold) was also altered by 11-KT. These genes encode conserved proteins that are related to the cell surface Ldl receptor, exhibit similar endocytosis functions, but also have fundamental roles in a diverse range of intercellular signal transduction pathways [45], interacting with multiple diverse ligands.

Pathway analysis also identified several potential canonical pathways and biological functions that further implicate 11-KT treatment with alterations in aspects of ovarian preparation for vitellogenesis and lipid uptake. The canonical pathways *insulin signaling* and *clathrin-mediated endocytosis signaling* were significantly associated with 11-KT treatment, supporting the previously discussed results. The biological function *concentration of lipids*, identified from the differential expression of 267 genes in our dataset, was predicted to be activated in 11-KT treated samples in comparison to controls, further supporting the hypothesis that 11-KT modulates lipid incorporation in the ovarian follicle. Similar results were observed from 11-KT treatment at the late perinucleolar stage of primary growth [4]. Clearly, 11-KT activates lipid transfer processes as evidenced by the current and previous studies across a range of teleost species.

**ii. 11-KT effects on Fsh signaling in the ovary.** The expression of several genes with known effects on Fsh synthesis or action was altered after three days of 11-KT treatment, including reduced expression of *inha*, *inhbb*, *cyp19a1*, and *amh*, and increased expression of *amhr2*. During secondary growth, Fsh secretion [8,46–48] and the ovarian response to Fsh stimulation increases [49,50], peaking during vitellogenesis. Ovarian *fshr* expression follows this pattern [9,35,51]. Fsh signaling through the Fshr modulates the expression of a number of genes during early secondary growth [48,52]. We previously showed that in primary follicles, 11-KT increases both *fshr* expression as well as a major downstream target of Fshr signaling, *cyp19a1* [4], which encodes the enzyme that converts T to E2. Additionally, plasma E2 was increased, and we hypothesized that 11-KT functions to prepare the ovarian follicle for Fsh mediated effects in secondary growth. Expression of *cyp19a1* is generally relatively low prior to vitellogenesis, and in vitro effects of Fsh on *cyp19a1* expression in previtellogenic salmon have not been reported, but Fsh increased *cyp19a1a* expression in vitellogenic follicles of rainbow

trout in vitro [53] and the temporal pattern of plasma E2 and ovarian *fshr* transcripts are well correlated [52]. However, in contrast to results from our studies in primary follicles, 11-KT had no effect on expression of *fshr*, and *cyp19a1* expression was reduced after 3-days of 11-KT treatment in early secondary follicles. The reason for these dissimilar effects on expression between these stages are unclear, but are perhaps due to stage-dependent changes in endocrine or paracrine feedback of 11-KT on the ovary, and/or alterations in intracellular pathways mediating the effects of 11-KT.

Levels of two transcripts for *inhibin alpha* subunit (*inha*) and *inhibin beta b* subunit (*inhbb*), which encode monomers of the heterodimeric inhibin B protein complex were decreased following 11-KT treatment. Expression patterns of inhibin subunits (which also encode the dimeric activin protein complex consisting of two *inhibin beta* subunits) tend to be higher during earlier stages of ovarian follicle development [54,55], while the *alpha* subunit, and thus mature inhibins, increase in response to Fsh later in development. This suggests mature activins are produced earlier, potentially regulating early follicular development [56] while inhibins begin to play a role as follicles shift to Fsh-responsiveness. Interpreting the impact of inhibins and activins is challenging because of complexity of their structure and limited information in fishes. Homo or heterodimers of either Inhba and/or Inhbb form activins that can stimulate Fsh, whereas heterodimers of Inha with Inhba or Inhbb form inhibins that inhibit Fsh. In non-teleosts *inha* and *inhbb* subunits appear to be estrogen responsive [57,58], and although E2 levels were not significantly increased by 11-KT in this study, paracrine actions of estrogens on *inha* and *inhbb* expression cannot be dismissed.

The expression of *amhr* was significantly increased while the expression of *amh* was significantly decreased by 11-KT in the current study, indicating androgenic modulation of the Amh signal in early secondary growth ovarian follicles. In female fish, the role of Amh is not clear, and although it has been linked to early ovarian development, very little experimental information exists regarding its specific actions. It is unknown if Amh serves a similar function in teleost ovarian follicle progression to that in mammals, where it is also expressed in ovarian granulosa cells and functions in limiting the progression of follicle development [59], and maintaining the primordial follicle pool reserve by repressing the FSH signal. Expression of *amh* in female teleosts has been detected in the ovary of multiple species [60], albeit at a lower level than in testis. Expression is primarily restricted to granulosa cells and in general *amh* is expressed in primary growth follicles, and expression increases in early secondary growth and during vitellogenesis.

**iii. Changes in the expression of genes encoding structural and functional proteins.**
Alterations in expression of several genes encoding extracellular matrix (ECM) proteins in response to 11-KT indicates that 11-KT may be involved in regulating the structure of the ovarian follicle, as we have reported for primary follicles [4]. As oocytes increase in volume during the progression through primary and secondary growth stages, the ovarian follicle layers also undergo numerous changes to maintain structural and biochemical support, and to increase communication between the components of the ovarian follicle. The ECM interacts with ovarian follicle cells to regulate gene expression, cell differentiation, and cellular growth [61,62], and changes in composition may alter growth factor or hormone access to the developing oocyte [62].

The expression of several collagen type IV isoforms was decreased while the expression of type VI isoforms was increased by 11-KT treatment. Collagen type IVs are primarily basement membrane components [63], while type VIs perform various cytoprotective functions in the ECM [64], including interaction with various membrane receptors involved in intracellular signaling. The expression of another gene, *prolyl 4-hydroxylase, transmembrane* (*p4htm*) was dramatically increased (>300-fold). The protein encoded by *p4htm* is a collagen P4h, which in

addition to its role in oxygen sensing, participates in post-translational folding of collagen polypeptides and is essential in basement membrane structure during development. [65]. Likewise, the expression of decorin (*dcn*) was increased by 11-KT. Dcn binds type-I collagens and plays a role in ECM assembly, but also cell cycle regulation and apoptosis [66]. The role of Dcn in the fish ovary is not well described, although transcript levels correlate with *fshr* expression, peak during vitellogenesis in coho salmon [52], and are regulated by Fsh [49].

Facilitating cell-to-cell signaling is a major function of the ECM, and biological functions in the category of cell-to-cell signaling were significantly activated by 11-KT in both primary [4] and secondary ovarian follicles (present study). Additionally, the expression of a number of genes encoding gap junction and tight junction associated proteins, including claudin isoforms, which coordinate cell signaling and membrane trafficking [67], was altered by 11-KT in the present study. In mammals, androgens are involved in regulating the expression of tight junction protein encoding genes in reproductive tissues [68–70]. This suggests that 11-KT, by altering the expression of claudins, may be involved in modulating cell signaling and membrane trafficking in the ovarian follicle cell layers. In our previous study, 11-KT altered many transcripts linked to the ECM, including those encoding numerous forms of collagen and laminin [4]. Together with the present study, this provides compelling evidence that 11-KT modulates the structure of the ECM and may enhance intrafollicular communication.

**iv. Changes in growth factor signaling.** The expression of several growth factor ligands and receptors was altered by 11-K: tgf-beta superfamily member ligands bmp7, gdf3, and tgfb3 transcript levels were decreased while receptors acvr2a, bmpr1a (alk3), tgfbr2, and tgfbr3 (betaglycan) were increased. The increase in receptor expression implies an increase in Tgf-beta signaling potential. Signaling through these receptors is known to control a wide range of cell processes and tissue homeostasis. The breadth of Tgf-beta superfamily member genes with altered expression following 11-KT treatment provides further evidence that androgens may be involved in mediating many cellular processes in the ovary: IPA$^®$ analysis identified numerous cellular process pathways in the ovary containing Tgf-beta superfamily ligands and receptors that were predicted to be significantly activated by 11-KT treatment.

The expression of growth factor receptors *igf1r* and *igf2r* was increased by 11-KT, implicating androgens in the modulation of intraovarian Igf signaling. Igfs have various effects on the teleost ovary, although the majority of studies have focused on the role Igf1 and Igf2 in secondary follicle steroid production and oocyte maturation [71] or the teleost specific gonadal Igf3 [72,73], a diverse array of Igf binding proteins also regulate signaling [74]. Both Igf1 and Igf2 bind Igf1r, and only Igf1r has been shown to activate signaling pathways [75]. As in mammals, Igf2r may function to attenuate signaling by binding and internalizing the Igf2 peptides which are trafficked to lysosomes for degradation [76]. In fish, the primary endocrine source of Igf1 and Igf2 is the liver, which synthesizes and releases them into circulation, although *igf1* mRNA has been detected in oocytes and ovarian follicle cells in previtellogenic carp [77]. The IgfI protein has been localized to ovarian granulosa cells of previtellogenic oocytes in several species [77–79], where it binds cognate receptors and has been implicated in paracrine/autocrine regulation in the ovary [80]. Both Igf1 and Igf1r have been implicated in previtellogenic ovarian development or function in several species [10,80]. Androgen treatment has also been shown to stimulate plasma IgfI levels in coho salmon [81]. Thus, the effects of 11-KT on Igf signaling could involve both endocrine paracrine/autocrine mechanisms.

## Conclusions

In this study, we provide further evidence that androgens play important roles in previtellogenic ovarian follicle development. A non-aromatizable androgen (e.g., 11-KT) cannot be

converted to E2. Thus, the 11-KT induced alterations in the ovarian transcriptome are due to androgen signaling, either directly or indirectly via actions on non-ovarian sites in the brain-pituitary-ovary axis. The expression of thousands of genes was altered by 11-KT treatment, across a variety of cellular processes. Importantly, specific increases in expression of vitellogenic machinery and alterations in Fsh signaling indicate that androgens control important aspects of the early secondary ovarian follicle phenotype.

The potency and relatively low circulating levels of 11-KT at this stage raises the question of the precise androgen signaling mechanisms in the ovary, particularly the identity of the endogenous androgen ligand. Ar isoforms in several fish species display differences in affinity for various androgens [82–84], and may mediate different physiological processes. In order to better understand androgen mediated effects on ovarian development, a comprehensive analysis of the ovarian and circulating levels of androgens (including 5α-reduced metabolites) and their receptor binding characteristics is necessary, as well as an analysis of the proteomic changes associated with androgen treatment. The potency of the non-aromatizable androgen 11-KT in the ovary at this stage, in concert with the very low endogenous plasma levels suggests an autocrine/paracrine androgen signaling mechanism.

## Supporting information

**S1 Table. The table lists ovarian contigs regulated by 11-KT after 3 days, identified by DESeq2 (basemean > 10), showing gene symbol (human ortholog), entrez gene name, Ensembl/UniProt/SwissProt accession, expr log ratio, expr P-value, number of Qiagen Ingenuity Pathway Analysis networks, type of protein encoded by gene, Entrez human id, Entrez mouse id, and Entrez rat id.**
(XLSX)

## Acknowledgments

The authors thank Abby Tillotson, Anna Bute, Fritzie Celino, and Erica Curles for their assistance with fish maintenance and sampling and Mollie Middleton, Jon Dickey, and Shelly Nance for technical support with steroid hormone measurements.

## Author Contributions

**Conceptualization:** Christopher Monson, Graham Young.

**Data curation:** Giles Goetz.

**Formal analysis:** Christopher Monson, Giles Goetz.

**Funding acquisition:** Kristy Forsgren, Penny Swanson, Graham Young.

**Investigation:** Christopher Monson.

**Methodology:** Christopher Monson, Giles Goetz.

**Project administration:** Christopher Monson.

**Supervision:** Penny Swanson, Graham Young.

**Writing – original draft:** Christopher Monson.

**Writing – review & editing:** Kristy Forsgren, Penny Swanson, Graham Young.

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
