## [Decision Letter · Decision Letter 0]

9 Oct 2023

PONE-D-23-28919In vivo treatment with a non-aromatizable androgen rapidly alters the ovarian transcriptome of previtellogenic secondary growth coho salmon (Onchorhynchus kisutch)PLOS ONE

Dear Dr. Monson,

Thank you for submitting your manuscript to PLOS ONE. After careful consideration, we feel that it has merit but does not fully meet PLOS ONE’s publication criteria as it currently stands. Therefore, we invite you to submit a revised version of the manuscript that comprehensively addresses the points raised during the review process.

We look forward to receiving your revised manuscript.

Kind regards,

Michael Schubert

Academic Editor

PLOS ONE

"The authors thank Abby Tillotson, Anna Bute, Fritzie Celino, and Erica Curles for their assistance with fish maintenance and sampling and Mollie Middleton, Jon Dickey, and Shelly Nance for technical support with steroid hormone measurements. This study was supported by National Science Foundation awards NSF-OISE-0914009, NSF IOS-0940765, and NSF IOS-1921746."

 "This study was supported by National Science Foundation awards NSF-OISE-0914009 awarded to KF, NSF IOS-0940765 awarded to GY and NSF IOS-1921746 awarded to GY. NSF.gov. The funders had no role in study design, data collection and analysis, decision to publish, or preparation of the manuscript."

Reviewers' comments:

Reviewer's Responses to Questions

**Comments to the Author**

1. Is the manuscript technically sound, and do the data support the conclusions?

Reviewer #1: Partly

Reviewer #2: Partly

2. Has the statistical analysis been performed appropriately and rigorously? 

Reviewer #1: I Don't Know

Reviewer #2: Yes

3. Have the authors made all data underlying the findings in their manuscript fully available?

Reviewer #1: No

Reviewer #2: Yes

4. Is the manuscript presented in an intelligible fashion and written in standard English?

Reviewer #1: Yes

Reviewer #2: Yes

5. Review Comments to the Author

Reviewer #1: The authors tried to identify the positive effects of 11-KT to early secondary ovarian growth in the coho salmon, the fish ovarian tissues were selected at the cortical alveolus stage by tissue section. Changes in the ovarian follicle transcriptome were tested using RNA-Seq followed by pathway analysis after three days of 11-KT treated.

The sample size was good (n= 3 /group * 2 groups), BUT the figures are not readable by its bad resolution, I do not know what caused this problem.

"Fish were lethally sampled after 3 days. At that time, fork length, body weight, and gonad weight were measured', I

do not think these somatic parameters will change only after 3 days of treatment.

"Histological screening eliminated any female that displayed overtly asynchronous ovarian stages or were not at the cortical alveolus stage", it is good, since to the defined phenotype is important to elucidate the genotype, but how to determine "similar cortical alveoli abundance" ? (No asterisks is indicated in figure 1).

"De novo assembly"? is there no reference genome from Salmonid to re-mapping the reads or contigs?

In "Table 1", what are "Additional reads included in backbone"?

"A total of 63,048 of these contigs (99.4%) were annotatable", and "Duplicate contigs were collapsed to the gene-ID level using the average expression value relative to controls", have the authors confirmed the presence of isoforms or redundant genes ? (the 4th-WGD in salmonids is significant on the presence of gene isoforms).

"Cluster analysis of differentially expressed contigs in 11-KT treated samples (Fig. 2C) ", there is no figure 2C!? please correct it.

From figure 2B, the significant variation on the intra-group FC has been observed, do the authors ever consider this variation to the result interpretation ? is it explained by zero-inflation ?

"Duplicate contigs were collapsed .... resulting in 3,853 genes with expression altered by 11-KT (P-adjusted ≤0.1)", how did the authors condense (or reduce dimension) the 3853 gene into figure 3s? is there any probability to confirm the results?

Please discuss the results from the IPA analysis, since the Neuregulin Signaling is also significant (z-score=3.18), why the authors ignore it ? the authors only discuss the canonical (classic) reproductive pathways (steroidogenesis, vitellogenesis and lipid uptake and processing, Fsh signaling, and tgfb superfamily members).

Reviewer #2: Primary growth, secondary growth, and maturation are the three developmental stages of oocytes in fish. Earlier published studies by the corresponding author suggest that during the late primary stage of ovarian follicle development in coho salmon, androgens are potent drivers of growth, and sustained release pellets of 11-KT implanted in vivo, advanced follicle growth and altered the ovarian transcriptome. Previous studies in coho salmon have shown that at early secondary stage, in vitro and in vivo treatment with 11-KT increased the size of ovarian follicles sooner than E2, but did not lead to formation of cortical alveoli. On the other hand, in vivo or in vitro treatment with E2 promoted the formation of cortical alveoli, increased plasma E2 levels, as well as Fsh transcript levels, indicating entry into secondary growth. Hence, it was concluded that 11-KT could be involved in initiating the entry of the ovarian follicles into secondary growth, since it was more efficient than E2 in increasing the size of ovarian follicles.

In this study, the authors have focused on the early secondary growth. Here, sustained release pellets containing 11-KT were implanted in female coho salmon that had ovaries at the cortical alveolus stage. Changes in the ovarian follicle transcriptome were determined after three days using RNA-Seq followed by pathway analysis. The fundamental difference between the previous study and this one is the absence/presence of cortical alveoli. Females with overtly asynchronous ovarian stages were eliminated. All ovaries displayed early to mid-cortical alveolus stage phenotype.

Comments

Frozen ovarian samples from control, 11-KT, and E2 treated females (N=3 per group) were selected for RNA-Seq analysis. I could not see any mention of E2 treated females anywhere else in the results related to this work.

Sequencing resulted in 1.6 billion total reads from 9 samples (Table 1) , , , The Table mentions additional reads included in backbone (1 to 3). Please clarify the source of these reads.

It is stated that the presence of cortical alveoli is a histological indicator of entry into secondary growth in coho salmon and that secondary growth is characterized by an increase in ovarian E2 synthesis via FSH signaling. However, in this work, even though plasma 11-KT level increased several folds in treated fish, serum E2 levels were unaltered. RNASeq results too show that 11-KT did not affect the expression of fshr, and reduced the expression of hsd3b, cyp19a1 (enzymes that catalyze steroid and E2 synthesis). RNASeq data shows that 11 KT treatment increased Vgtr, while E2 was shown to repress vtgr expression in previtellogenic follicles of largemouth bass and medaka.

In my opinion, these major observations do not support the conclusion that ‘after three days of sex steroid exposure,. . . ., the widespread transcriptomic changes induced by 11-KT are consistent with a role for 11-KT in lipid and vitellogenin uptake and processing, and in Fsh signaling, which are hallmarks of secondary growth, as well as cellular development and other cellular processes, and changes in the extracellular matrix. Hence, I would suggest more pragmatic inference of the data.

6. PLOS authors have the option to publish the peer review history of their article (what does this mean?). If published, this will include your full peer review and any attached files.

Reviewer #1: **Yes: **Yung-Sen Huang

Reviewer #2: No

---

## [Author Response · Author response to Decision Letter 0]

22 Jan 2024

Reviewer #1: The authors tried to identify the positive effects of 11-KT to early secondary ovarian growth in the coho salmon, the fish ovarian tissues were selected at the cortical alveolus stage by tissue section. Changes in the ovarian follicle transcriptome were tested using RNA-Seq followed by pathway analysis after three days of 11-KT treated.

The sample size was good (n= 3 /group * 2 groups), BUT the figures are not readable by its bad resolution, I do not know what caused this problem.

We refer to the editors if the figures do not match required resolution standards. We are not sure why they may be appearing unreadable in the document viewed by reviewer #1.

"Fish were lethally sampled after 3 days. At that time, fork length, body weight, and gonad weight were measured', I

do not think these somatic parameters will change only after 3 days of treatment.

We agree that these somatic parameters would be highly unlikely to change after just three days of treatment but included these data to indicate that the fish were of the same size and weight and therefore likely to be in the same stage of development since coho salmon are semelparous.

"Histological screening eliminated any female that displayed overtly asynchronous ovarian stages or were not at the cortical alveolus stage", it is good, since to the defined phenotype is important to elucidate the genotype, but how to determine "similar cortical alveoli abundance" ? (No asterisks is indicated in figure 1).

We did not quantify cortical alveoli abundance but did use observations of cortical alveoli as an indication of developmental stage. We selected females that displayed the most similar morphological phenotypes and that were clearly in the ECA stage of development.

"De novo assembly"? is there no reference genome from Salmonid to re-mapping the reads or contigs?

The assembly was done prior to the coho salmon genome being published.

In "Table 1", what are "Additional reads included in backbone"?

We initially sequenced E2 treated females of the same cohort and those reads were included in the initial backbone that the treatments were then compared back to. However, we chose to focus our analysis in this manuscript on the 11-KT treatment group out of a concern for brevity. We added text to the figure legend to clear this up.

"A total of 63,048 of these contigs (99.4%) were annotatable", and "Duplicate contigs were collapsed to the gene-ID level using the average expression value relative to controls", have the authors confirmed the presence of isoforms or redundant genes? (the 4th-WGD in salmonids is significant on the presence of gene isoforms).

We were able to identify isoforms of many genes when sequence information was available. However, for pathway analyses, we could only use genes recognized by the software and thus were limited in our inclusion of salmonid-specific isoforms to those shared by zebrafish. One of the requirements of the pathway analysis software is the collapse of duplicate entries into single gene-level IDs. In doing this we lost potential isoforms that are not shared by zebrafish or homo sapiens as the case may be.

"Cluster analysis of differentially expressed contigs in 11-KT treated samples (Fig. 2C) ", there is no figure 2C!? please correct it.

2C should have been labelled 2B, apologies and thank you for catching that error

From figure 2B, the significant variation on the intra-group FC has been observed, do the authors ever consider this variation to the result interpretation? is it explained by zero-inflation ?

The authors are unsure what the reviewer is asking with this question. There is indeed differences in the level of differential expression between individual genes in the treatment groups as would be expected in a full-transcriptome analysis.

"Duplicate contigs were collapsed .... resulting in 3,853 genes with expression altered by 11-KT (P-adjusted ≤0.1)", how did the authors condense (or reduce dimension) the 3853 gene into figure 3s? is there any probability to confirm the results?

In figure 3 we are highlighting specific genes that were altered and that we discuss throughout the manuscript, we are not including all of the genes identified. Table S1 has the complete list of genes along with the Entrez, Uniprot, SwissProt accession numbers and ids. The traditional way of confirming RNA-Seq results is by qPCR, but we believe that there are more variables and thus less reliability in the application of qPCR than in the RNA-Seq the method used here, and so qPCR would be unlikely to give accurate confirmation of expression patterns across all of the genes we discuss.

Please discuss the results from the IPA analysis, since the Neuregulin Signaling is also significant (z-score=3.18), why the authors ignore it ? the authors only discuss the canonical (classic) reproductive pathways (steroidogenesis, vitellogenesis and lipid uptake and processing, Fsh signaling, and tgfb superfamily members).

Due to the enormity of the dataset and our relative expertise in reproductive development, we chose to focus our analysis on only a subset of the genes and pathways that were significantly altered by the treatment. We agree that this dataset opens many potentially interesting hypotheses. There is abundant evidence that the EGF and EGF-like family members play important roles in development of many tissues, with NRGs playing particular roles in granulosa cell survival, but to our knowledge of the current literature, studies are almost exclusively in mammalian models. We include a minor number of mammalian references where appropriate but attempted to keep our discussion within the teleost literature as much as possible.

Reviewer #2: Primary growth, secondary growth, and maturation are the three developmental stages of oocytes in fish. Earlier published studies by the corresponding author suggest that during the late primary stage of ovarian follicle development in coho salmon, androgens are potent drivers of growth, and sustained release pellets of 11-KT implanted in vivo, advanced follicle growth and altered the ovarian transcriptome. Previous studies in coho salmon have shown that at early secondary stage, in vitro and in vivo treatment with 11-KT increased the size of ovarian follicles sooner than E2, but did not lead to formation of cortical alveoli. On the other hand, in vivo or in vitro treatment with E2 promoted the formation of cortical alveoli, increased plasma E2 levels, as well as Fsh transcript levels, indicating entry into secondary growth. Hence, it was concluded that 11-KT could be involved in initiating the entry of the ovarian follicles into secondary growth, since it was more efficient than E2 in increasing the size of ovarian follicles.

In this study, the authors have focused on the early secondary growth. Here, sustained release pellets containing 11-KT were implanted in female coho salmon that had ovaries at the cortical alveolus stage. Changes in the ovarian follicle transcriptome were determined after three days using RNA-Seq followed by pathway analysis. The fundamental difference between the previous study and this one is the absence/presence of cortical alveoli. Females with overtly asynchronous ovarian stages were eliminated. All ovaries displayed early to mid-cortical alveolus stage phenotype.

Comments

Frozen ovarian samples from control, 11-KT, and E2 treated females (N=3 per group) were selected for RNA-Seq analysis. I could not see any mention of E2 treated females anywhere else in the results related to this work.

Removed text that mentions E2 treated females from the methods section.

Sequencing resulted in 1.6 billion total reads from 9 samples (Table 1) , , , The Table mentions additional reads included in backbone (1 to 3). Please clarify the source of these reads.

We initially sequenced E2 treated females of the same cohort and those reads were included in the initial backbone that the treatments were then compared back to. However, we chose to focus our analysis in this manuscript on the 11-KT treatment group out of a concern for brevity. We added text to the figure legend to clear this up

It is stated that the presence of cortical alveoli is a histological indicator of entry into secondary growth in coho salmon and that secondary growth is characterized by an increase in ovarian E2 synthesis via FSH signaling. However, in this work, even though plasma 11-KT level increased several folds in treated fish, serum E2 levels were unaltered. RNASeq results too show that 11-KT did not affect the expression of fshr, and reduced the expression of hsd3b, cyp19a1 (enzymes that catalyze steroid and E2 synthesis). RNASeq data shows that 11 KT treatment increased Vgtr, while E2 was shown to repress vtgr expression in previtellogenic follicles of largemouth bass and medaka.

In my opinion, these major observations do not support the conclusion that ‘after three days of sex steroid exposure,. . . ., the widespread transcriptomic changes induced by 11-KT are consistent with a role for 11-KT in lipid and vitellogenin uptake and processing, and in Fsh signaling, which are hallmarks of secondary growth, as well as cellular development and other cellular processes, and changes in the extracellular matrix. Hence, I would suggest more pragmatic inference of the data.

We respectfully disagree with the reviewer on this point. 11-KT induced changes in late primary growth (previous study cited throughout) included increased expression of fshr and cyp19a1. Combined with these results in early secondary growth, especially alterations of Fsh signaling pathway transcripts, the increase in expression of lipid and vitellogenin uptake genes, and genes in the steroidogenic pathways, as well as the general predictions of the pathway analysis, we believe that androgens likely play an important role in preparing the follicle for the events that occur later in secondary growth, and in particular Fsh-mediated processes.

---

## [Decision Letter · Decision Letter 1]

21 Feb 2024

PONE-D-23-28919R1In vivo treatment with a non-aromatizable androgen rapidly alters the ovarian transcriptome of previtellogenic secondary growth coho salmon (Onchorhynchus kisutch)PLOS ONE

Dear Dr. Monson,

Thank you for submitting your manuscript to PLOS ONE. After careful consideration, we feel that it has merit but does not fully meet PLOS ONE’s publication criteria as it currently stands. Therefore, we invite you to submit a revised version of the manuscript that addresses the points raised during the review process.  Please submit your revised manuscript by Apr 06 2024 11:59PM. If you will need more time than this to complete your revisions, please reply to this message or contact the journal office at plosone@plos.org. Please include the following items when submitting your revised manuscript:A rebuttal letter that responds to each point raised by the academic editor and reviewer(s). You should upload this letter as a separate file labeled 'Response to Reviewers'.A marked-up copy of your manuscript that highlights changes made to the original version. You should upload this as a separate file labeled 'Revised Manuscript with Track Changes'.An unmarked version of your revised paper without tracked changes. You should upload this as a separate file labeled 'Manuscript'.

We look forward to receiving your revised manuscript.

Kind regards,

Michael Schubert

Academic Editor

PLOS ONE

**Additional Editor Comments: **

The reviewer notes severe shortcomings concerning the quality of the manuscript's figures. These shortcomings have to be resolved in full before the manuscript can be considered for publication in PLOS ONE.

Reviewers' comments:

Reviewer's Responses to Questions

**Comments to the Author**

1. If the authors have adequately addressed your comments raised in a previous round of review and you feel that this manuscript is now acceptable for publication, you may indicate that here to bypass the “Comments to the Author” section, enter your conflict of interest statement in the “Confidential to Editor” section, and submit your "Accept" recommendation.

Reviewer #1: All comments have been addressed

2. Is the manuscript technically sound, and do the data support the conclusions?

Reviewer #1: Partly

3. Has the statistical analysis been performed appropriately and rigorously? 

Reviewer #1: I Don't Know

4. Have the authors made all data underlying the findings in their manuscript fully available?

Reviewer #1: Yes

5. Is the manuscript presented in an intelligible fashion and written in standard English?

Reviewer #1: Yes

6. Review Comments to the Author

Reviewer #1: The resolution of the figures is still too low, they are not clear to provide the informations, and even something is missed in the figure 3.

7. PLOS authors have the option to publish the peer review history of their article (what does this mean?). If published, this will include your full peer review and any attached files.

Reviewer #1: No

---

## [Author Response · Author response to Decision Letter 1]

19 Sep 2024

Reviewer #1: The authors tried to identify the positive effects of 11-KT to early secondary ovarian growth in the coho salmon, the fish ovarian tissues were selected at the cortical alveolus stage by tissue section. Changes in the ovarian follicle transcriptome were tested using RNA-Seq followed by pathway analysis after three days of 11-KT treated.

The sample size was good (n= 3 /group * 2 groups), BUT the figures are not readable by its bad resolution, I do not know what caused this problem.

We refer to the editors if the figures do not match required resolution standards. We are not sure why they may be appearing unreadable in the document viewed by reviewer #1.

"Fish were lethally sampled after 3 days. At that time, fork length, body weight, and gonad weight were measured', I

do not think these somatic parameters will change only after 3 days of treatment.

We agree that these somatic parameters would be highly unlikely to change after just three days of treatment but included these data to indicate that the fish were of the same size and weight and therefore likely to be in the same stage of development since coho salmon are semelparous.

"Histological screening eliminated any female that displayed overtly asynchronous ovarian stages or were not at the cortical alveolus stage", it is good, since to the defined phenotype is important to elucidate the genotype, but how to determine "similar cortical alveoli abundance" ? (No asterisks is indicated in figure 1).

We did not quantify cortical alveoli abundance but did use observations of cortical alveoli as an indication of developmental stage. We selected females that displayed the most similar morphological phenotypes and that were clearly in the ECA stage of development.

"De novo assembly"? is there no reference genome from Salmonid to re-mapping the reads or contigs?

The assembly was done prior to the coho salmon genome being published.

In "Table 1", what are "Additional reads included in backbone"?

We initially sequenced E2 treated females of the same cohort and those reads were included in the initial backbone that the treatments were then compared back to. However, we chose to focus our analysis in this manuscript on the 11-KT treatment group out of a concern for brevity. We added text to the figure legend to clear this up.

"A total of 63,048 of these contigs (99.4%) were annotatable", and "Duplicate contigs were collapsed to the gene-ID level using the average expression value relative to controls", have the authors confirmed the presence of isoforms or redundant genes? (the 4th-WGD in salmonids is significant on the presence of gene isoforms).

We were able to identify isoforms of many genes when sequence information was available. However, for pathway analyses, we could only use genes recognized by the software and thus were limited in our inclusion of salmonid-specific isoforms to those shared by zebrafish. One of the requirements of the pathway analysis software is the collapse of duplicate entries into single gene-level IDs. In doing this we lost potential isoforms that are not shared by zebrafish or homo sapiens as the case may be.

"Cluster analysis of differentially expressed contigs in 11-KT treated samples (Fig. 2C) ", there is no figure 2C!? please correct it.

2C should have been labelled 2B, apologies and thank you for catching that error

From figure 2B, the significant variation on the intra-group FC has been observed, do the authors ever consider this variation to the result interpretation? is it explained by zero-inflation ?

The authors are unsure what the reviewer is asking with this question. There is indeed differences in the level of differential expression between individual genes in the treatment groups as would be expected in a full-transcriptome analysis.

"Duplicate contigs were collapsed .... resulting in 3,853 genes with expression altered by 11-KT (P-adjusted ≤0.1)", how did the authors condense (or reduce dimension) the 3853 gene into figure 3s? is there any probability to confirm the results?

In figure 3 we are highlighting specific genes that were altered and that we discuss throughout the manuscript, we are not including all of the genes identified. Table S1 has the complete list of genes along with the Entrez, Uniprot, SwissProt accession numbers and ids. The traditional way of confirming RNA-Seq results is by qPCR, but we believe that there are more variables and thus less reliability in the application of qPCR than in the RNA-Seq the method used here, and so qPCR would be unlikely to give accurate confirmation of expression patterns across all of the genes we discuss.

Please discuss the results from the IPA analysis, since the Neuregulin Signaling is also significant (z-score=3.18), why the authors ignore it ? the authors only discuss the canonical (classic) reproductive pathways (steroidogenesis, vitellogenesis and lipid uptake and processing, Fsh signaling, and tgfb superfamily members).

Due to the enormity of the dataset and our relative expertise in reproductive development, we chose to focus our analysis on only a subset of the genes and pathways that were significantly altered by the treatment. We agree that this dataset opens many potentially interesting hypotheses. There is abundant evidence that the EGF and EGF-like family members play important roles in development of many tissues, with NRGs playing particular roles in granulosa cell survival, but to our knowledge of the current literature, studies are almost exclusively in mammalian models. We include a minor number of mammalian references where appropriate but attempted to keep our discussion within the teleost literature as much as possible.

Reviewer #2: Primary growth, secondary growth, and maturation are the three developmental stages of oocytes in fish. Earlier published studies by the corresponding author suggest that during the late primary stage of ovarian follicle development in coho salmon, androgens are potent drivers of growth, and sustained release pellets of 11-KT implanted in vivo, advanced follicle growth and altered the ovarian transcriptome. Previous studies in coho salmon have shown that at early secondary stage, in vitro and in vivo treatment with 11-KT increased the size of ovarian follicles sooner than E2, but did not lead to formation of cortical alveoli. On the other hand, in vivo or in vitro treatment with E2 promoted the formation of cortical alveoli, increased plasma E2 levels, as well as Fsh transcript levels, indicating entry into secondary growth. Hence, it was concluded that 11-KT could be involved in initiating the entry of the ovarian follicles into secondary growth, since it was more efficient than E2 in increasing the size of ovarian follicles.

In this study, the authors have focused on the early secondary growth. Here, sustained release pellets containing 11-KT were implanted in female coho salmon that had ovaries at the cortical alveolus stage. Changes in the ovarian follicle transcriptome were determined after three days using RNA-Seq followed by pathway analysis. The fundamental difference between the previous study and this one is the absence/presence of cortical alveoli. Females with overtly asynchronous ovarian stages were eliminated. All ovaries displayed early to mid-cortical alveolus stage phenotype.

Comments

Frozen ovarian samples from control, 11-KT, and E2 treated females (N=3 per group) were selected for RNA-Seq analysis. I could not see any mention of E2 treated females anywhere else in the results related to this work.

Removed text that mentions E2 treated females from the methods section.

Sequencing resulted in 1.6 billion total reads from 9 samples (Table 1) , , , The Table mentions additional reads included in backbone (1 to 3). Please clarify the source of these reads.

We initially sequenced E2 treated females of the same cohort and those reads were included in the initial backbone that the treatments were then compared back to. However, we chose to focus our analysis in this manuscript on the 11-KT treatment group out of a concern for brevity. We added text to the figure legend to clear this up

It is stated that the presence of cortical alveoli is a histological indicator of entry into secondary growth in coho salmon and that secondary growth is characterized by an increase in ovarian E2 synthesis via FSH signaling. However, in this work, even though plasma 11-KT level increased several folds in treated fish, serum E2 levels were unaltered. RNASeq results too show that 11-KT did not affect the expression of fshr, and reduced the expression of hsd3b, cyp19a1 (enzymes that catalyze steroid and E2 synthesis). RNASeq data shows that 11 KT treatment increased Vgtr, while E2 was shown to repress vtgr expression in previtellogenic follicles of largemouth bass and medaka.

In my opinion, these major observations do not support the conclusion that ‘after three days of sex steroid exposure,. . . ., the widespread transcriptomic changes induced by 11-KT are consistent with a role for 11-KT in lipid and vitellogenin uptake and processing, and in Fsh signaling, which are hallmarks of secondary growth, as well as cellular development and other cellular processes, and changes in the extracellular matrix. Hence, I would suggest more pragmatic inference of the data.

We respectfully disagree with the reviewer on this point. 11-KT induced changes in late primary growth (previous study cited throughout) included increased expression of fshr and cyp19a1. Combined with these results in early secondary growth, especially alterations of Fsh signaling pathway transcripts, the increase in expression of lipid and vitellogenin uptake genes, and genes in the steroidogenic pathways, as well as the general predictions of the pathway analysis, we believe that androgens likely play an important role in preparing the follicle for the events that occur later in secondary growth, and in particular Fsh-mediated processes.

---

## [Editor Report · Decision Letter 2]

23 Sep 2024

In vivo treatment with a non-aromatizable androgen rapidly alters the ovarian transcriptome of previtellogenic secondary growth coho salmon (Onchorhynchus kisutch)

PONE-D-23-28919R2

Dear Dr. Monson,

We’re pleased to inform you that your manuscript has been judged scientifically suitable for publication and will be formally accepted for publication once it meets all outstanding technical requirements.

Kind regards,

Michael Schubert

Academic Editor

PLOS ONE

---

## [Editor Report · Acceptance letter]

30 Sep 2024

PONE-D-23-28919R2 

PLOS ONE

Dear Dr. Monson, 

I'm pleased to inform you that your manuscript has been deemed suitable for publication in PLOS ONE. Congratulations! Your manuscript is now being handed over to our production team.

Kind regards, 

on behalf of

Dr. Michael Schubert 

Academic Editor

PLOS ONE